# Dietary Interventions for Healthy Pregnant Women: A Systematic Review of Tools to Promote a Healthy Antenatal Dietary Intake

**DOI:** 10.3390/nu12071981

**Published:** 2020-07-03

**Authors:** Yvette H. Beulen, Sabina Super, Jeanne H.M. de Vries, Maria A. Koelen, Edith J.M. Feskens, Annemarie Wagemakers

**Affiliations:** 1Health and Society, Social Sciences Group, Wageningen University & Research, 6706 KN Wageningen, The Netherlands; sabina.super@wur.nl (S.S.); maria.koelen@wur.nl (M.A.K.); annemarie.wagemakers@wur.nl (A.W.); 2Division of Human Nutrition and Health, Wageningen University & Research, 6708 WE Wageningen, The Netherlands; jeanne.devries@wur.nl (J.H.M.d.V.); edith.feskens@wur.nl (E.J.M.F.)

**Keywords:** pregnancy, health promotion tools, nutrition

## Abstract

Maternal nutrition is essential for the development and lifelong health of the offspring. Antenatal care provides unique opportunities for nutrition communication, and health promotion tools (e.g., guidelines, instruments, packages, or resources) might help to overcome several concurrent barriers. We conducted a systematic literature review to map tools that are available for the promotion of a healthy dietary intake in healthy pregnant women in Western countries, and to identify what makes these tools feasible and effective for these women and their healthcare providers. Seventeen studies were included, evaluating tools with various delivery modes, content, and providers. Nine studies employed multiple, complementary delivery methods and almost all studies (n = 14) tailored the content to varying degrees, based on the individual characteristics and lifestyle behaviors of the participants. We found that the feasibility of a tool was dependent on practical issues, time investment, and providers’ motivation, skills, and knowledge, while the effectiveness was related more to the type of provider and the content. Most effective interventions were provided by dietitians and nutritionists, and were highly tailored. Based on the results of this review, we believe that custom tools that are sensitive to inequalities are needed to support all women in obtaining or maintaining a healthy diet during pregnancy.

## 1. Introduction

Maternal nutrition is of great importance for fetal development and growth, as well as offspring health, throughout the life course [1]. A healthy antenatal dietary intake supports fetal development, and might thereby prevent congenital malformations, premature birth, and low birth weight. Although pregnant women are generally aware of the importance of a healthy diet during pregnancy, their actual dietary intake remains sub-optimal [2,3,4]. In particular, women with a lower socioeconomic status (SES) adhere less to dietary guidelines and have poorer maternal and child health outcomes than women from a more privileged socioeconomic background [5].

Pregnancy is often regarded as an ideal time for improving dietary intake, with most opportunities in antenatal care. Increased interest in nutrition [6], women’s trust in healthcare providers [7], and regular antenatal visits are considered facilitators of nutrition communication in antenatal care [8,9]. However, several barriers have to be overcome for antenatal care providers (e.g., midwives, obstetricians, and GP’s) to integrate nutrition communication into their daily practice. The main barriers include limited self-perceived nutrition expertise and self-efficacy by providers, and a lack of time [10,11,12]. Considering the communication with low SES pregnant women, specifically, providers might experience language barriers, and complex needs within and outside the scope of antenatal care (e.g., housing, welfare) might be prioritized over discussing a healthy diet [13,14].

Evidence-based health promotion tools might provide a solution to address some of these barriers, successfully integrate nutrition communication into antenatal care, and take away some of these barriers. Based on a definition of health promotion tools by McCalman et al. [15], we define ‘tools’ as any structured guideline, instrument, package or resource that supports antenatal care providers in integrating nutrition communication into their current practice. We also consider (financial) resources and education materials directed at pregnant women as tools, since these might also save healthcare providers (HCPs) time, facilitate nutrition communication, and improve women’s dietary intake. For example, tools assessing diet quality might be used to complement counseling sessions by increasing awareness of dietary intake and motivating women to eat healthier or to provide support in dietary self-monitoring and advice [16,17]. 

To better implement health promotion tools in practice, insight into their feasibility and effectiveness is needed. Literature on the feasibility and effectiveness of antenatal nutrition interventions has been reviewed numerous times, yet most studies focused on gestational weight gain (GWG) in overweight and obese populations [18,19,20,21], gestational diabetes mellitus (GDM), or other clinical pregnancy and birth outcomes, such as (pre)eclampsia and preterm birth [22,23,24,25,26,27,28,29]. This focus might be due to guidelines and recommendations by various national and international health policy organizations such as WHO [30] and NICE [31], which often discuss gestational weight gain and preventing non-communicable diseases, rather than promoting a healthy diet in general. Moreover, we noticed that only one prior review (also on gestational weight gain in overweight and obese pregnant women) evaluated not only the type of dietary intervention but also delivery methods, content, and dietary intake [21]. Gaining insight into intermediary outcomes, as well as the characteristics of successful interventions, is essential to unravel the mechanisms through which interventions might improve dietary intake, and ultimately the health outcomes.

The current review addresses two important gaps in the current literature—(1) an overview of tools to promote healthy dietary behavior in healthy pregnant women of all BMI categories, and (2) insights into characteristics of tools that make them feasible for recipients and providers, which are effective at improving dietary intake. Therefore, the aim of this review was to provide an overview of tools that could be integrated into antenatal care for promotion of healthy dietary behavior in healthy pregnant women, and to gain insights into the characteristics of those tools that make them feasible and effective.

## 2. Materials and Methods 

The reporting of this systematic literature review was guided by the PRISMA statement [20], as well as the Cochrane Handbook for Systematic Reviews [32]. Reference management software Mendeley Desktop version 1.19.5 was used during the screening procedure, and Microsoft Excel 2016 was used in the data extraction phase.

### 2.1. Search Strategy

A literature search for journal articles was conducted in two electronic databases—PubMed and Web of Science. PubMed was chosen because it uses the MEDLINE database, and includes up-to-date citations that are not yet indexed, records from journals that are not indexed, and records considered ‘out-of-scope’ from journals that are partially indexed for this database. Additionally, Web of Science was recommended by the librarian, to minimize selection bias.

A PICO model was used to formulate the search strategy [33] (see Appendix A). The population (P) of interest was healthy pregnant women, receiving antenatal care in Western countries. The Intervention (I) was defined as any tool for nutrition communication that could be used by HCP, or complement antenatal care. Included studies did not necessarily need to have a control group (C). If they did have a control group, the control group should receive standard antenatal care, or a less intensive form of nutrition communication than the intervention. Outcomes (O) of interest should at least include maternal dietary intake or behavioral determinants of dietary intake. 

The search strategy was adapted to each database and included keywords and Medical Subject Headings (MeSH) terms. Initially, we tried including search terms for low SES populations and for healthcare providers in the search strategy, but we removed those terms as they provided insufficient studies to conduct a review. The full electronic search strategies for both databases are listed in Appendix B. The databases were last searched September 2019. 

We extended our search through backward snowballing and searching trial registries. Reference lists of, both, included studies and the relevant systematic reviews identified, were searched. The ClinicalTrials.gov and WHO International Clinical Trials Registry Platform (ICTRP) portal were searched for inventions with a ‘completed’ status in February 2020.

### 2.2. Eligibility Criteria

To be included, records had to be journal articles describing at least one tool with the potential of being used by healthcare providers, to promote a healthy dietary intake in healthy pregnant women. Following the removal of duplicates, titles, and abstracts were screened by two independent reviewers (Y.B., S.S.) against seven initial exclusion criteria. First, the study population had to be healthy pregnant women. Animal studies and studies in a non-pregnant population (e.g., focusing on breastfeeding and child feeding practices, or preconception), or a diseased pregnant population (i.e., women with GDM, hyperemesis gravidarum, preeclampsia, hypertension, and pre-existing chronic conditions) were excluded. Second, studies had to describe one or multiple tool(s) aimed at promoting a healthy dietary intake in this population. Studies not describing any tool or tools with another objective than to promote a healthy dietary intake (e.g., focus on gestational weight gain or glycemic control, physical activity, and dental health) were excluded. Micronutrient supplementation trials without any behavior change element were also excluded based on this criterion. Third, articles from non-Western countries were excluded, where we defined the Western world as Europe, the Americas, and Australasia. We also excluded (4) studies describing tools that could not be used by healthcare professionals (e.g., focused on a workplace setting and governmental financial aid programs) or (5) outdated tools (e.g., based on outdated devices such as a Personal Digital Assistant). Last, we excluded (6) records with no full text available (including conference abstracts), or (7) those only available in language other than English or Dutch. Disagreement on the inclusion of a study was resolved through consensus or the consultation of a third reviewer (A.W.). 

### 2.3. Quality Appraisal

Quality appraisal was performed by two independent researchers (Y.B., S.S.), using checklists developed at the Joanna Briggs Institute for RCTs, quasi-experimental studies, and qualitative studies [34]. These checklists guided the appraisal of strengths and weaknesses of the available evidence. The reviewers further discussed the magnitude of flaws, relevancy, and applicability issues, and whether the results and conclusions were supported by the data. Studies were then classified as having a low, moderate, or high risk of bias, based on the number and severity of flaws. Flaws were classified as minor or major, where examples of minor flaws were minor details missing or inadequately reported, unclear tables or figures, or no explanations for drop-out rates. Examples of major flaws were poor justification for conducting the study, questionable reliability of data collection method, or unjustified conclusions. In studies classified as low risk, only few minor flaws were identified. Those classified as moderate risk contained one or two major flaws or several minor flaws. Studies perceived as high-risk, contained multiple major flaws and were consequently excluded from this review (see Figure 1). 

### 2.4. Data Extraction

General characteristics (authors, title, year, and place of publication), study aim, design, and population (number of participants, age, parity, gestational age, and indicators of socioeconomic status), details of the tool(s) used (type, content, and theoretical approach), and relevant outcome measures were extracted by one reviewer (Y.B.), checked by a second (S.S.), and discussed with all authors.

The outcomes of interest in this review were feasibility of tools for both recipients and providers, and effectiveness of tools at improving dietary intake. As most studies did not specifically mention any feasibility outcome, we conceptualized relevant outcomes and outcome measures, based on common process evaluation concepts [35,36]. We considered feasibility outcomes related to recruitment and retention, as well as acceptability, according to participants and providers. It was thereby essential to distinguish between the feasibility of tools and research activities. For example, we extracted retention rates of intervention activities, but not of data collection timepoints. Table 1 provides a conceptualization of the outcomes of interests, based on the literature on designing evaluation studies, as well as an overview of related outcome measures.

### 2.5. Data Synthesis

We separated data on delivery mode, content, provider, and frequency of delivery and duration of the intervention to break down important elements contributing to the feasibility and effectiveness of interventions. Most studies combined several tools in their intervention; thus, the outcome measures represent the effectiveness and, partly, the feasibility of the complete intervention rather than the feasibility of individual tools. Where possible, we provided outcomes, such as response rates, acceptability, and dose received for the separate tools applied in the intervention.

As pregnant women in low SES populations adhere less to dietary guidelines, we were especially interested in the needs of these women. We, therefore, described results for low SES populations separately wherever possible.

Data were analyzed through a narrative synthesis, drawn from tables and taking into account biases and other issues potentially affecting interpretation of each study’s findings, as identified by the critical appraisal process [37]. 

## 3. Results

Seventeen papers met the inclusion criteria and were considered of sufficient quality after critical appraisal. The flow diagram of the study selection is provided in Figure 1. Ten studies had a moderate risk of bias, and seven studies had a low risk. Twelve studies were randomized controlled trials, of which, one used a 5 × 2 factorial design, and five studies were formative evaluations. Most studies used quantitative data collection methods, one RCT study used mixed methods and two formative evaluations used only qualitative methods.

Table 2 shows the main characteristics of the included studies and study populations. The studies were conducted in Australia (*n* = 6) [39,41,43,51,53,54], the USA (*n* = 5) [38,45,47,48,49], and across Europe (*n* = 6), including the UK (*n* = 3) [38,40,52], Norway [44], Finland (*n* = 1) [50], and Greece (*n* = 1) [46]. The five studies from the USA all included a majority of Hispanic participants. Participants in three of these studies [47,48,49], as well as those in two studies from Greece [46] and the UK [40], were disadvantaged in terms of education, employment or income, at either the individual or community level. 

Participants were most commonly recruited at antenatal clinics or community health centers. In a majority of studies, women were recruited at booking visits (*n* = 4), or a maximum gestational age (*n* = 8) was defined to recruit them in early pregnancy. Thus, most participants were enrolled in the studies in their first or second trimester, often between 14- and 20-weeks of gestation. Three studies recruited participants from pre-existing RCTs on antenatal nutrition [41], influenza vaccination [49], or mother and infant nutrition and probiotics [50].

Behavior change models were commonly used in designing the intervention or questionnaires. In eleven studies, the authors specified using behavior change and decision-making theories and frameworks, including the Transtheoretical Model (or Stages of Change) [32,33,40], the Health Belief Model [42,49], Social Cognitive Theory [42], Self Determination Theory [52], and Theory of Reasoned Action model [38]. Other studies included Motivational Interviewing [45] and the 5As model [53,54], to guide behavior change conversations. In the remaining six studies, no use of theoretical frameworks was specified. 

### 3.1. Types of Tools and Their Use

A first analysis of the studies selected demonstrated the diversity of tools that were employed to improve the dietary behavior of pregnant women (see Table 3).

#### 3.1.1. Delivery Mode

Approximately half of the studies (*n* = 9) employed multiple, complementary delivery modes. Mobile Health (mHealth) tools and printed materials were most commonly used, but in different ways. The mHealth tools were mostly used independently, while the printed materials were used as an adjunct to more intensive delivery modes, such as face-to-face counseling, in six out of eight studies using these materials. Hillesund et al. and Dodd et al. provided the most comprehensive interventions in terms of delivery modes, combining telephone calls, mHealth tools, and printed materials with face-to-face consultations [41], or a cooking class [44].

#### 3.1.2. Content

Content of the interventions was often tailored, based on individual characteristics and lifestyle behaviors of participants, and varied across studies. In total, fourteen studies tailored the intervention in some way, although the extent to which the advice was tailored, varied. Tailoring was either achieved through counseling or personal goalsetting guided by providers [39,41,44,50,51,52], through self-assessment/screening, or self-monitoring [38,43,45,47,48,54].

Besides nutrition content, the majority of interventions (*n* = 12) included content related to other lifestyle behaviors related to pregnancy or the postnatal period, such as physical activity, smoking, breastfeeding, sleep, and emotional wellbeing. Nutrition content ranged from overall dietary behavior to specific foods or food groups. Interventions covering overall dietary behavior addressed aspects of nutrition, such as limiting the consumption of energy-dense foods and increasing consumption of nutrient-dense foods, based on national dietary guidelines [39,41,44,45,51,52,54]. Four studies specifically focused on increasing the fruit and vegetable intake [40,42,47,49], of which one study promoted the consumption of fruit juices [40]. Some studies extended the dietary advice by addressing issues such as serving sizes [39,44,52], food preparation [44,48], micronutrient supplementation [39,40], or recommended weight gain [45,54]. 

#### 3.1.3. Providers

Tools were delivered by various providers, but most commonly by dietitians and nutritionists. Extensive face-to-face and telephone counseling or video feedback was often provided by dietitians [39,41,51] or nutritionists [44,50], but also public-health nutrition students [44], trained nurses [46], researchers trained in motivational interviewing [52], and exercise physiologists [51], who provided such counseling. Midwives generally played a less intensive role. They were mainly involved in distributing printed information [38,40,54] and providing basic face-to-face advice [40].

#### 3.1.4. Timing, Frequency, and Duration

Intervention intensity, including frequency and duration, varied within and between delivery modes. The intensity ranged from a single printed resource provided at the booking visit [54] to face-to-face counseling sessions every 2 weeks from enrollment (mean gestational age 17 weeks) [46]. The majority of interventions started around women’s first antenatal care visit and included at least one follow-up. In most studies including counselling, sessions were scheduled 4 to 6 weeks apart [41,44,51]. 

### 3.2. Feasibility 

Measures of feasibility, such as reach, response rates, and retention rates, varied widely between studies (Table 4). While one study completed and even exceeded recruitment goals within a matter of weeks instead of the planned months [48], another experienced severe issues with recruitment and retention [51]. We found that participants in some studies conducted in disadvantaged populations [40,42,47,48] were more likely to retain participants than those conducted in higher SES populations [51,53,54]. To take Rissel et al. as an example, a first challenge was meeting the inclusion criterion of ≤18 weeks gestation, as a substantial proportion of women (mainly in rural hospitals) did not visit the hospital this early in pregnancy. A subsequent challenge in this study was retaining the women who agreed to participate. Of 923 women recruited, 322 women enrolled (27.6%), and 89 women completed the final call (9.6%). There were no significant differences in retention between the health coaching or information-only group [51].

#### 3.2.1. Delivery Mode

Feasibility of different delivery modes for participants depended mostly on the required time investment, practical issues, and interest in the topic. In several studies, women mentioned a longer intervention duration and lack of time as reasons to decline participation or withdraw from the intervention [45,50,51,53]. We also observed that three of the studies with low retention rates were among the most intensive, with up to eight contact moments (see Table 2). On the other hand, Wilkinson and McIntyre conducted a low intensity, one-off group workshop, and reported that about half (48.3%) of intervention participants attended this workshop. Women who provided feedback for their nonattendance, mentioned practical problems with accessing a large, inner city hospital (especially parking) and getting time off work to attend a workshop [53]. The main reason for declining participation in the study by Rissel et al. was non-interest in managing weight, which was one of the aims of the intervention besides promoting a healthy diet and physical activity [51]. 

A combination of delivery modes or different tools using the same delivery mode also seemed to enhance feasibility. For example, participants in some studies liked that they received a printed resource after a counseling session [45,52], or complementary mHealth resources [43]. The study by Warren et al. provides a nice example of the value of the printed resource in addition to face-to-face and via telephone counseling. All participants in this study responded positively to the individualized goal card they received and reported that they had referred back to it. For some, the goal card acted as a reminder, and ensured them that they had achieved their goals [52]. In addition, as one participant described it, women also appreciated having ‘something in black and white’. Ashman et al. found that participants in their study thought the combination of a visual video summary and a detailed telephone consultation with a dietitian was helpful, and that a summary alone would not have been enough [39]. The participants also perceived text-messages as a helpful reminder to record dietary intake. 

#### 3.2.2. Content

Women wanted to receive credible information and appreciated tools that were tailored and practical. They expressed positive experiences with tools that allowed them to set personalized dietary goals [41,52]. Participants in the qualitative study by Warren et al. said that the counseling session incorporating motivational interviewing and individual goal setting made them re-assess their eating and think differently about their diet, and gave them a sense of reassurance [52]. In another study, participants considered practical information such as portion size, food groups and recipe suggestions to be particularly useful [41].

#### 3.2.3. Providers

Providers play a crucial role with regard to the feasibility of an intervention program. Rissel et al. found that women withdrawing upon enrolment initially agreed to participate primarily because they were asked by their midwife [51]. Participants in the study by Ashman et al. thought a telephone consultation with a dietitian was more detailed and easier to understand than the visual video summary they received and therefore was a valuable addition [39].

The feasibility of tools for providers was hardly addressed in any of the included studies. Of all included studies, only Rissel et al. conducted interviews with a convenience sample of midwives (N = 19) and practitioners (N = 5). These midwives were generally positive about the program. Mauriello et al. (2011) evaluated the ability of staff to recruit women. The staff easily exceeded the anticipated sample due to the willingness of staff and the eagerness of pregnant women attending the health center. The staff also remarked that they found the program easy to implement within the prenatal care flow and that participation did not influence or disrupt the delivery of prenatal care within the health centers. 

#### 3.2.4. Timing, Frequency, and Duration

Multiple contact moments might be useful to avoid information overload, and could help to spread time investment for both women and providers, across pregnancy. Participants in two different studies with one counseling or mHealth session remarked that the session was too long [45,48]. Both studies provided printed materials for participants and healthcare providers, which might have been helpful as reminders (Section 3.2.1).

### 3.3. Effectiveness

Effectiveness of studies varied widely (Table 4). Outcome measures of effectiveness were related to dietary intake and other health behaviors, or on determinants of these behaviors. Sixteen studies included dietary behavior as an outcome, although a range of data collection methods was used. Methods included—(validated) questions integrated into general questionnaires (*n* = 8) [42,45,47,48,49,51,53,54], food frequency questionnaires (*n* = 3) [39,41,44], (weighed) food records (*n* = 3) [38,39,50], biomarkers of exposure (*n* = 2) [40,46], 24-h dietary recalls (*n* = 1) [39], and interviews (*n* = 1) [52]). Seven studies included information on behavioral determinants (i.e., nutritional knowledge [38,45], attitudes [38,42], information-seeking behaviors [44,45], intentions [47,48], and beliefs [48,49]).

Approximately half of the included studies (*n* = 8) showed positive effects on dietary behavior. Another two studies showed improvements in beta-carotene concentrations, serum vitamin C concentrations and fruit juice intake, but did not find significant effects on the other micro nutrients of interest [46] or in fresh fruit intake [40]. Six studies showed no significant improvements in dietary behavior. Half of these studies did find improvements in behavioral determinants, such as nutrition knowledge [38], attitudes towards alcohol (in higher educated women) [42], and progression to action or maintenance Stages of Changes, according to the Transtheoretical model [47]. Three of the eight studies that found improvements in dietary behavior also reported that nutritional knowledge, attitudes, and beliefs had improved [45,48,49]. Several studies also reported increased eating occasions [39], or increases in the number of servings of specific food groups, e.g., dairy [41] or healthy snacks [44]. 

#### 3.3.1. Delivery Mode

Overall, effectiveness of the interventions seemed to rely on the content and the providers, to a greater extent, than on the mode of delivery. All effective studies included mHealth [39,45,47,48,49], counseling [50], or a group activity [53], or a combination of those [44]. On the other hand, an equal number of studies included (a combination of) those delivery modes, not showing any effect. Dodd et al. found that the addition of a smartphone app was not associated with any significant difference in intakes, compared with lifestyle advice alone [41]. The two studies solely using printed materials did not improve dietary behavior [38,54], although one of those studies did find an increase in nutritional knowledge [38].

#### 3.3.2. Content

Seven out of eight effective interventions included at least one tailored component. For example, the Video Doctor program by Jackson et al. could match video clips to participant’s BMI, eating and exercise habits, and readiness to change through in-depth, digital behavioral risk assessments, a database of counseling video clips, and extensive branching logic. Participants in the intervention group obtained more nutritional knowledge and discussed nutrition more often with their HCP. The authors also observed statistically significant increases in intakes of fruits and vegetables, whole grains, fish, avocado and nuts, and significant decreases in intake of sugary foods, refined grains, high-fat meats, fried foods, solid fats, and fast food [45].

Behavior change models were commonly used in designing the interventions or questionnaires and were effective in the majority of studies, but not all. Six of the effective studies designed their interventions or tailored the content based on behavior-change theories, including the Transtheoretical model (Stage of Change) [47,48], the Health Belief model [49], or the Self Determination Theory [52]. Two effective studies (additionally) used behavior change techniques/models to guide communication, such as motivational interviewing [45,52] and the 5As model [53]. However, four other studies using these same or similar theoretical frameworks did not show any effects [38,41,42,54].

#### 3.3.3. Providers

Most effective studies were provided by a dietitian or nutritionist (*n* = 4), although there were two other interventions provided by (amongst others) dietitians that were not found to be effective [41,51]. In the effective Video Doctor program, an ‘ideal conversation with a health care provider’ was simulated through a Video Doctor actor. ‘Real’ HCPs were involved only after they received a printed cueing sheet including women’s risk profiles and the suggested counseling statements [45]. 

#### 3.3.4. Timing, Frequency, and Duration

A long, intensive intervention did not necessarily lead to improvements in dietary behavior. Dodd et al. and Rissel et al. provided very comprehensive interventions with many counseling sessions, but did not find any significant improvements in dietary behavior [41,51]. In contrast, participants in the study of Wilkinson and McIntyre improved their intakes, after ‘only’ receiving a booklet and attending a 60-min cooking workshop [53].

## 4. Discussion

### 4.1. Main Findings, Interpreted in Perspective of Previous Studies

The aim of this review was to provide an overview of tools to promote healthy dietary behavior in all pregnant women and to gain insight into the characteristics of those tools that make them feasible and effective. The overview showed that most interventions were complex and included multiple tools. The available tools and their use varied widely between studies. The most commonly used delivery modes were the Mobile Health (mHealth) tools and the printed materials. The content was tailored based on individual characteristics and behaviors, and nutrition was addressed amongst other lifestyle behaviors in a majority of interventions. Providers were not always described but were most often dietitians and nutritionists.

Although we aimed to address a wide range of tools, our overview included relatively many mHealth tools. About half of the included studies (8/17) involved some kind of mHealth, which showed the abundance of such tools in recent years. Although the internet and mobile applications are popular sources of information for pregnant women, previous studies have found the quality of information provided through such resources is doubtful and that the apps often contain limited pregnancy-specific nutrition information [43,55,56]. In several studies, women expressed a clear preference for quality assured online resources recommended by a trusted health professional [43,57]. 

We found that the feasibility of a tool depended on time investment and practical issues related to the delivery mode, as well as on providers’ motivation, skills, and knowledge. A combination of delivery modes or different tools using the same delivery mode seemed to enhance feasibility. Reminders and summaries worked particularly well for many participants. Previous studies also suggested that written advice might influence knowledge about nutrition, but that other forms of nutrition communication are needed to achieve changes in behavior and attitudes [38]. 

The effectiveness of tools to promote healthy dietary behavior is likely to depend primarily on the type of provider and on the content. Most effective interventions included in this review were provided by dietitians and nutritionists and were highly tailored. Previous reviews and meta-analyses on tailored online information have argued that tailored messages are more effective in bringing about behavior change than static information [58,59]. Our findings suggest that tailored content might contribute to but does not guarantee success. Although almost all studies in our review (*n* = 14) included one or multiple tools with tailored nutrition content, only half of these studies were successful at improving dietary behavior or behavioral determinants. This might be explained by the approach and degree to which interventions were tailored (e.g., through counseling by HCP versus self-assessment).

Wherever possible, we retrieved information on a low SES population specifically. Five studies in this review included women from disadvantaged populations. Low SES populations are often considered ‘hard to reach’, yet they are most likely to benefit the most from nutrition interventions. Interestingly, we found that participants in some studies conducted in disadvantaged populations [40,42,47,48] were more likely to retain participants than those conducted in higher SES populations. This might be explained (partly) by the intensity and delivery of the tools, which required a minimal time investment and were either delivered at home or prior to an antenatal visit. Moreover, three studies with very successful recruitment and retention were conducted in mainly Hispanic populations in USA [47,48,49]. It might be that these studies were more tailored to this specific population (e.g., culturally tailored) than studies reporting more trouble in reaching and retaining women, which were often conducted in women at all SES levels or relatively high SES, in Australia [51,53,54].

Based on the included studies, we could not fully disseminate which specific characteristics of tools contributed to the feasibility and effectiveness. Most studies combined multiple tools and did not evaluate the tools separately. Although the overall quality of the included studies was good, the results of this review, therefore, need to be treated with some caution.

### 4.2. Strengths and Limitations

This review is the first to provide a detailed overview of the (characteristics of) available tools to promote healthy dietary behavior in healthy pregnant women. Notable strengths of the conduct and reporting of this review are the comprehensive search strategy, the use of independent researchers for screening and critical appraisal, and compliance with the PRISMA statement. The search strategy was guided by our objective and the PICO model, and reviewed by a qualified librarian. Creating the search string was an iterative process, in which especially various (synonyms of) potential tools were continuously added and removed, to strike a balance between striving for comprehensiveness and maintain relevance. Unlike other systematic literature reviews [59,60,61] we did not search only for mHealth tools, although they were included as potential tools in our search string through MeSH terms (see Appendix B). 

As the quality of a systematic review depends largely on the quality of the papers included, we were very critical of which studies to include throughout the process of writing this review. Finding an appropriate quality appraisal tool was challenging, as most tools are very suitable for designs such as randomized controlled trials but are limited in appraising other types of research. We ultimately chose to use checklists by the Joanna Briggs Institute, which are available for a range of study designs. We excluded studies with high risk of bias, according to the critical appraisal. 

A main limitation of this review was the heterogeneity of the included studies. Despite strict eligibility criteria, the studies were very heterogeneous with regard to study design, population, objectives, methods, and outcomes. As such, no meta-analysis could be performed, data extraction, quality appraisal and data synthesis were challenging, and the results of this review should be treated with caution. Inclusion and exclusion of articles was not always straightforward. For example, we aimed to include only studies focused on healthy dietary behavior rather than on gestational weight management, but in some cases it was difficult to determine the main focus of an article in the screening process. Therefore, some of the included studies still discussed appropriate weight gain as an aim [44,51]. Nevertheless, we believe these studies still made valuable contributions to the current systematic literature review, as they also discussed promoting healthy dietary behavior.

While we excluded studies using outdated modes of delivery, we did not exclude studies based on the topicality of the dietary recommendations. As a result, some studies used rather outdated dietary recommendations, e.g., Anderson et al. (macronutrient EN% guidelines, low-fat diet) [38], Burr et al. (fruit juice) [40], and Kafatos et al. (snacks) [46]. However, the inclusion of these studies did provide useful insights into the aspects of tools other than content, such as the cost of their intervention [40].

We only partially succeeded in our aim to examine the feasibility and effectiveness of tools. Data available in the retrieved papers were limited with regard to various outcome measures, or the results could not be linked to one specific tool. An important limitation was the inability to assess feasibility of tools for HCP, based on the included studies, as most studies were mainly focused on recipients. Process evaluation measures related to providers, such as ‘dose delivered’ and ‘fidelity’ were hardly addressed in any of the included studies. Furthermore, many interventions used multiple tools, which were not evaluated independently. This made it impossible to compare the measures of effectiveness and of feasibility, across different tools. 

Ideally, we would have had full insight into the contextual factors, such as organizational, community, social, and political factors underlying the included studies. Strategic communication strategies to promote the program to both women and staff within organizations are essential for successful adoption from planning through to evaluation [62]. Based on what is reported in the papers, however, we know very little about how tools were provided exactly and how they were communicated to both women and staff. One study found that women were susceptible to their midwife’s offer to participate, but apparently the midwives were unable to convince them to remain in the study [51]. Reasons provided by women declining participation or withdrawing after enrollment included a lack of time and non-interest. Explanations from the perspective of providers are lacking. However, a potential reason for low retention rates might be a lack of continuity of care. Care models might vary widely within and between countries, and could lead to considerable variation in continuity of care and time available [63]. The grey literature was not addressed in the current systematic literature review but could provide more insights in considering tools for a specific context.

### 4.3. Recommendations for Future Research

Based on the results of our review, we could only conclude that one size did not fit all. Specifically related to low income populations, we agreed with Barker et al.’s recommendation that ‘we need both to help women to feel more in control of their food choices and to make it easier for them to make better choices’ [64]. We would like to add to this that, in turn, researchers and policy makers must support the HCP in gaining confidence and by collaboratively developing tools that fit within the current practice. As shown in this review, the feasibility of tools, and therefore the active and sustained use of tools in practice, strongly depends on the motivation and skills of providers. Currently, we know little about the needs of providers and about potential barriers and facilitators to providing nutrition communication. It is, therefore, essential to assess whether antenatal care providers see a role for themselves in nutrition communication, and whether they could be trained to provide nutrition communication, or if a consultation with a dietitian or nutritionist could be, for instance, effectively integrated into antenatal care. Basu et al. provided preliminary evidence for the first option and showed that a compact training model to assist practicing midwives in providing (amongst others) nutrition communication increased midwives’ knowledge and confidence [65]. Best-practice, however, should always be determined on the basis of resources available at a specific setting.

The overview of tools provided by this review could help HCP or intervention developers choose an appropriate tool for their setting. Using the results, we plan to develop both a toolbox and a full strategy to be implemented in the Netherlands. Through this review, we identified various opportunities for integrating nutrition communication into Dutch antenatal care. First of all, a suite of delivery modes might be helpful to improve feasibility for both providers and recipients [53]. As Seward et al. previously suggested, “interventions should be personalized not only in the approach and content, but also to the women’s preferences in mode of communication and technological tools to support goals tracking” [66]. Furthermore, we strongly believe in the potential of evidence-based mHealth tools, not only in providing information, as well as screening, goalsetting, goal-tracking, and counselling. A major benefit of mHealth tools is their accessibility, and future generations will likely continue to use technology to learn, track, and communicate.

## 5. Conclusions

Custom tools that identify and address inequalities in health are needed to enhance health equity across generations. Various tools are available to promote a healthy dietary intake in all pregnant women, not only those overweight or obese. For those tools to be both feasible and effective, they should be easily accessible, include tailored advice, and preferably be provided by a dietitian or nutritionist. In particular, mHealth tools in combination with other delivery modes could help to integrate such nutrition communication into antenatal care in an effective, efficient, and sustainable manner.

## Figures and Tables

**Figure 1 nutrients-12-01981-f001:**
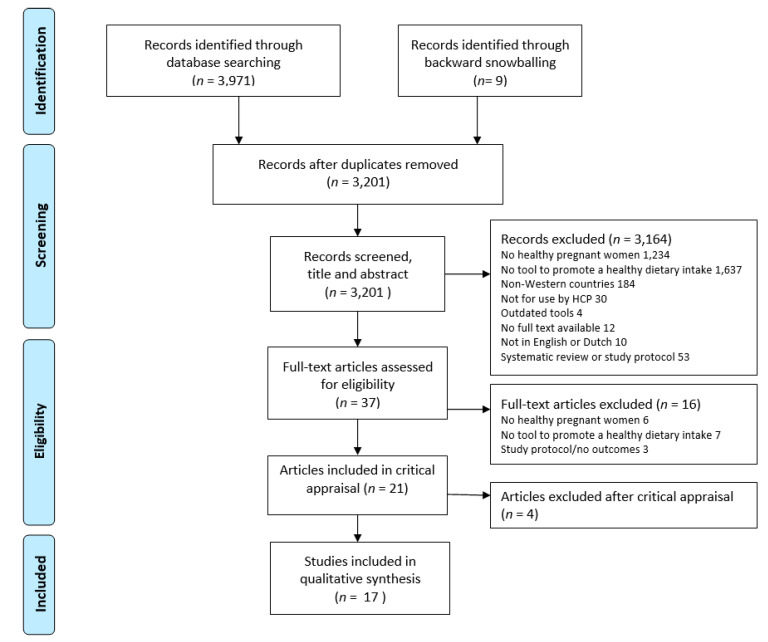
Flowchart of study selection.

**Table 1 nutrients-12-01981-t001:** Overview outcomes and outcome measures of interest.

Outcome	Outcome Measures
Feasibility	Recruitment & retention: Reach: “The proportion of intended target audience that participates in the intervention” [35] Response rate Retention rate
	Acceptability: ‘Participant’s satisfaction with the program and interactions with staff or investigators’ [35] Dose received: ‘The extent to which participants actively engage with, interact with, are receptive to, or use materials or recommended resources. It is a characteristic of the target audience that assesses the extent of engagement of participants with the intervention’ [36]
Effectiveness	Behavioral determinants: Nutritional knowledge Awareness Attitude Self-efficacy Intention Information-seeking behavior Beliefs
	Dietary behavior: Dietary intake Diet quality Nutritional status

**Table 2 nutrients-12-01981-t002:** Overview of bibliographic information, objectives, study design and population, and risk of bias in the included studies.

Authors (Year)	Country	Objective	Study Design	Population: Ethnicity, SES Indicators, BMI	N	Risk of Bias
Anderson et al. (1995) [38]	UK	To test the response of pregnant women to dietary advice by comparing the nutrition knowledge, attitudinal variables to healthier eating and nutrient intake in a group of women receiving routine antenatal clinical dietary education and a group who also received a special intervention education program.	RCT	All SES levels	I: 141; C: 145	Low
Ashman et al. (2016) [39]	Australia	To assess the relative validity of the SNAQ tool for analyzing dietary intake, compared by the nutrient analysis software, to describe the nutritional intake adequacy of pregnant participants, and to assess acceptability of dietary feedback via smartphone.	Formative evaluation	All born in Australia, 31% of Indigenous descent; mostly higher educated (54% university degree)	27	Moderate
Burr et al. (2007) [40]	UK	To examine the effectiveness of the two methods of increasing fruit and fruit juice intake in pregnancy—midwives’ advice and vouchers exchangeable for juice.	RCT	Lower SES population, attending antenatal clinic in a deprived area	190	Moderate
Dodd et al. (2018) [41]	Australia	To evaluate the impact of a smartphone application as an adjunct to standard face-to-face consultations in facilitating dietary and physical activity change among pregnant women with BMI ≥ 18.5 kg/m2.	RCT (nested)	Participants of two pregnancy nutrition-based RCTs; majority Caucasian (73%); all SES levels; overall 42.6% normal weight, 19.1% overweight, 38.3% obese	I1: 77; I2: 85	Low
Evans et al. (2012) [42]	USA	To assess audience exposure, awareness, and cognitive and affective reactions to text4baby messages; and to identify direct effects of text4baby messages on maternal pre-natal care and related health attitudes, beliefs, and behavioral outcomes, and related health-promoting and risk-avoidance behaviors.	RCT	Majority Hispanic (80%); randomly sampled from a largely low-income population, participants mostly lower educated (76% ≤ High School education)	123	Moderate
Hearn et al. (2014) [43]	Australia	To determine online information needs of perinatal women regarding healthy eating, physical activity, and healthy weight during pregnancy and the first eighteen months postpartum.	Formative evaluation (qualitative)	Majority higher SES levels	2378	Moderate
Hillesund et al. (2016) [44]	Norway	To investigate whether a lifestyle intervention during pregnancy offering supervised exercise groups and simplified dietary advice would optimize pregnancy weight gain and provide measurable health effects for mother and newborn.	RCT	All levels of education and household income; mean BMI intervention group 23.8 (4.1), control group 23.5 (3.7)	508	Low
Jackson et al. (2011) [45]	USA	To determine if an interactive, computerized Video Doctor counselling tool improves self-reported diet and exercise in pregnant women.	RCT	Majority Hispanic (39%) or African–American (24%); mostly higher educated (52% college and above); overall mean BMI 27.0, 44% overweight/obese	I: 163; C: 164	Low
Kafatos et al. (1989) [46]	Greece	To assess dietary habits and the impact of nutrition education among pregnant women in the rural county of Florina, Northern Greece.	RCT	Majority lower SES (70–73%); mean BMI Intervention group 23.1 (0.2), control group 22.7 (0.2)	I: 300; C: 268	Moderate
Mauriello et al. (2016) [47]	USA	To test an iPad-delivered multiple behavior tailored intervention (Healthy Pregnancy: Step-by-Step) for pregnant women that address smoking cessation, stress management, and fruit and vegetable consumption.	2 x 5 factorial design	Majority Hispanic (65%); mostly lower educated (68% ≤ High School education), 51% unemployed	I: 169; C: 166	Moderate
Mauriello et al. (2011) [48]	USA	To promote positive health behaviors during pregnancy among a low-income population, across multiple ethnic groups.	Formative evaluation	Majority Hispanic (46%) or white, non-Hispanic (32%); mostly lower educated (78% ≤ High School)	87	Moderate
Moniz et al. (2015) [49]	USA	To delineate the effects of texts messages sent to pregnant women to promote preventive health beliefs and behaviors.	RCT (nested)	Participants of an RCT to improve influenza vaccination rates; majority black (61%); mostly lower educated (80% ≤ High School) and lower household incomes	171	Moderate
Piirainen et al. (2016) [50]	Finland	To assess the impact of dietary counseling, combined with the provision of food products on food and nutrient intake in pregnant women.	RCT (nested)	Participants of a mother and infant nutrition and probiotic study; relatively high educated (47-52% university or college); 12% underweight, 61% normal weight; 21% overweight, 7% obese	209	Moderate
Rissel et al. (2019) [51]	Australia	To compare the outcomes of the two Get Healthy in Pregnancy (GHiP) options, to determine the characteristics of women likely to use the service and to explore the feedback from women and health professionals.	RCT (clustered, mixed methods)	Ethnicity and SES indicators not reported; intervention group 60.2% overweight/obese, control group 50.3% overweight/obese	I: 180; C: 146	Moderate
Warren et al. (2017) [52]	UK	To assess the feasibility and acceptability of the ‘Eat Well Keep Active’ intervention program designed to promote healthy eating and physical activity in pregnant women.	Formative evaluation (qualitative)	All Caucasian; 90% employed (60% skilled); 70% normal weight, 30% overweight (obese women were excluded)	20	Low
Wilkinson & McIntyre (2012) [53]	Australia	To deliver a low-intensity, dietitian-led behavior change workshop at a Maternity Hospital, to influence behaviors with demonstrated health outcomes.	RCT	Women attending a Tertiary maternal health service; 99% non-indigenous; relatively high educated (39–43% degree/higher degree), majority employed (69-76%) and high household incomes; mean BMI Intervention group 25.4 (5.2), control group 24.6 (5.5)	I: 178; C: 182	Low
Wilkinson et al. (2010) [54]	Australia	To evaluate the effectiveness of a women-focused, woman-held companion to usual obstetric care (the ‘Pregnancy Pocketbook’) for improving smoking cessation, fruit and vegetable intake, and PA, during pregnancy.	Formative evaluation	95–97% non-indigenous; relatively many participants did not finish high school (24–33%), 47–55% employed, relatively many women with full time home duties (36–46%); majority high household incomes; mean BMI Intervention group 25.7 (6.0), control group 25.0 (5.7)	I: 140; C: 130	Low

**Table 3 nutrients-12-01981-t003:** An overview of interventions, tools, and characteristics of tools.

Reference	Intervention Name	Delivery Mode	Description	Provider	Timing & Frequency	Dietary Guidance
Anderson et al. (1995) [38]	Food for life	Written	Pack 1: Self-assessment quiz, information booklet, and shopping list pad; Pack 2: Personalized letter from a named doctor, recipe leaflet	Midwife	At inclusion and 26 weeks gestation	Specific food recommendations identified by examining the food selections in women with a high-fat intake compared to those with a low-fat intake
Ashman et al. (2016) [39]	Diet Bytes	mHealth	Image-based dietary assessment through Evernote app, training on how to use the app to record dietary intake, feedback via video	Dietitian	Dietary assessment weeks 1–4, feedback weeks 6	Personalized content, including core and energy-dense, nutrient-poor food groups and intakes of selected nutrients, practical tailored examples of foods and serving sizes
Burr et al. (2007) [40]		Face-to-face, written, foods	Advice group: Advice and written information (leaflet), Voucher group: Received vouchers to be exchanged for free cartons of pure fruit juice	Midwife	2L of fruit juice/weeks for 30 weeks	Focused on increasing the amount of fruit and fruit juice in pregnant women’s diet
Dodd et al. (2018) [41]	SNAPP trial	Face-to-face, telephone, mHealth, written	Interactive smartphone application as an adjunct to standard face-to-face consultations, telephone calls, and written materials	Dietitian, research assistant	Face-to-face within 2 weeks of entry and at 28 weeks gestation, telephone at 22, 24, and 32 weeks, written at 36 weeks gestation	Dietary advice consistent with the Australian Guide to Healthy Eating—balance of macronutrients, reduced intake of foods high in refined carbohydrates and saturated fats, increased intake of fiber and of fruit, vegetables, and dairy
Evans et al. (2012) [42]	Text4baby	mHealth	Text messaging service, designed to build self-efficacy, knowledge, and skills	Not reported	Not reported	Fruit and vegetable intake, vitamin supplementation, alcohol
Hearn et al. (2014) [43]	Healthy You, Healthy Baby	mHealth	Website to provide women with convenient access to brief factual information, and an accompanying app with a self-assessment tool to track lifestyle behaviors and weight	Not reported	Highest self-assessment usage in first 2 trimesters	Individualized content (nutrition and weight)
Hillesund et al. (2016) [44]	Norwegian Fit for Delivery	Telephone, written, mHealth, group activity	A pamphlet on dietary recommendations, telephone sessions incorporating a woman’s own experience of which aspects of their diet and dietary behavior needed improvement, a cooking class and access to a password-protected website with recipes and practical tips on cooking	Nutritionist, nutrition students	Pamphlet soon after entry, two telephone sessions scheduled 4–6 weeks apart, one-evening cooking class	Ten dietary recommendations targeting energy balance, fruit and vegetable intake, consumption of water vs. sweetened beverages, and frequency and portion size of non-core foods
Jackson et al. (2011) [45]	Keep fit (Video Doctor)	Written, mHealth	Computer program delivered on laptops in clinic, including in-depth behavioral risk assessments, tailored counselling messages, and printed output for women and clinicians	Video-doctor actor	10–15 min assessment, follow-up at least 4 weeks later	Individualized content focused on increasing intake of fruits, vegetables, and whole grains, increasing healthful versus unhealthful fats and decreasing sugary foods; weight gain
Kafatos et al. (1989) [46]		Face-to-face	Face-to-face nutrition counselling through home visits by trained nurses	Trained nurses	Home visits every 2 weeks	Not reported
Mauriello et al. (2011) [48]	Healthy Pregnancy: Step by Step	mHealth	Computer-based modules addressing self-selected behaviors, including messages and feedback on stages of change, decisional balance, self-efficacy, and processes of change	Not reported	One-off, during wait for booking visit	Basics of nutrition during pregnancy, including food sources and methods for selecting a balanced diet, practical techniques, consumption of locally grown foods that have a high nutrient value and food preparation and preservation to reduce the loss of nutrients
Mauriello et al. (2016) [47]	Healthy Pregnancy: Step by Step	mHealth, written	Tailored iPad-delivered intervention consisting of interactive sessions focused on two self-selected health behavior risks (see above), and a printed multiple behavior change guide	Not reported	Approximately 25 min before regular antenatal visits, printed guide at first session	Focused on increased fruit and vegetable consumption, written materials address nutrition and healthy eating more globally (exact content unclear)
Moniz et al. (2015) [49]		mHealth	Text messages about general preventive health measures in pregnancy	Not reported	12 weekly text messages	Individualized content focused on fruit and vegetable consumption
Piirainen et al. (2016) [50]		Face-to-face, foods	Detailed dietary counselling and provision of conventional food products with favorable fat and fiber content for use at home	Nutritionist	Visits at each trimester	Daily vitamin use, dietary discretion
Rissel et al. (2019) [51]	Get Healthy in Pregnancy (GHiP)	Telephone, written	Evidence-based written resources plus a journey booklet to record progress and health coaching calls	Various HCP (e.g., dietitians, exercise physiologists)	Start of both arms between 12 and 22 weeks gestation. Information only arm: one 20–30 min call, telephone-based coaching arm: up to 8 calls	Focused on the amount and the type of fat and the amount of fiber in the diet, consumption of fruits and vegetables, wholegrain bread and cereals, leaner meat products, low-fat cheese and milk products, vegetable oil or soft margarine, and fish
Warren et al. (2017) [52]	Eat Well Keep Active	Face-to-face, telephone, written	A brief counselling session incorporating motivational interviewing and individual goal setting, personalized magnetic goal card, and follow-up telephone call	Researcher trained in MI	10–15 min Counselling session at approximately 16 weeks gestation, goal card sent within a week, 5 min telephone call two weeks after initial session	Dietary advice consistent with Australian Guide to Healthy Eating, recommended weight gain during pregnancy, micronutrients (e.g., folate, iodine, iron), foods to avoid, portion sizes and serves, healthy plate and food labels
Wilkinson & McIntyre (2012) [53]	Healthy Start to Pregnancy	Group activity, written	Workshop (capacity 15 women, +/- partners), including screening tools, information and behavior change strategies and links to more specialized services, and a healthy eating during pregnancy booklet	Dietitian	Booklet at their booking visit, one 60 min workshop session	Individualized content
Wilkinson et al. (2010) [54]	Pregnancy Pocketbook	Written	Interactive resource, with evidence-based information, screening tools, goal setting and self-monitoring activities, and referral information.	Midwife	Pocketbook delivered at booking visit	Fruit and vegetable intake, fat, fiber and overall diet quality, healthy weight gain

**Table 4 nutrients-12-01981-t004:** Feasibility and effectiveness outcomes [39].

Reference	Feasibility		Effectiveness	
	*Recruitment and Retention*	*Acceptability*	*Behavioral Determinants*	*Dietary Behavior*
Anderson et al. (1995) [38]	Not reported	Not reported	Nutritional knowledge (particularly practical applications) higher in the intervention group. No significant differences for attitude variables.	No significant differences in micronutrient intakes and energy composition
Ashman et al. (2016) [39]	92% recorded dietary intake on all 3 days	96% thought the combination of a video summary (‘visual’) and a follow-up telephone consultation with a dietitian (‘detailed’, ‘easier to understand’) was helpful	Not reported	77% of participants in the final survey reported changing their diet (foods or food groups, nutrient intakes, or eating behaviors) and some switched to healthier cooking methods
Burr et al. (2007) [40]	190 out of 192 women invited agreed to participate.	Of the 37 participants who still received juice at 32 weeks, all claimed to drink it, although 25 shared it (mostly with children or partners). The main barrier to consumption was change in taste and appetite, followed by the perishability of fruit.	Not reported	A significant increase of fruit juice intake and serum β-carotene, but no increase in consumption of fresh fruits.
Dodd et al. (2018) [41]	Not reported	31% reported using the smartphone app; 50% of users liked the smartphone app (the other 50% provided no response, or answered ‘undecided’) and found the information useful, particularly practical and recipe suggestions, portion size, food groups, and goalsetting opportunities.	Not reported	No significant differences in macronutrient and food group intakes between smartphone and advice vs. advice only.
Evans et al. (2012) [42]	400,000 individuals enrolled in the service between launch and publication of the article, 73% retention rate	Not reported	No differences in attitudes regarding fruit and vegetable consumption, or micronutrient supplementation. Attitudes towards alcohol consumption improved in higher educated participants.	No significant improvements in fruit and vegetable intake.
Hearn et al. (2014) [43]	2378 users signed up to the app over the first year, which is 7% of the target group and 18 % first time mothers in WA. Antenatal web pages were viewed 14,023 times. Usage was highest in the first two trimesters and postpartum.	Website pages with nutrition content were viewed more (40% of views) than the pages on weight, physical activity, sleep, emotions and social life, but self-assessment on sleep and weight were more popular in the app. The average person completed 3.6 self-assessment questionnaires, 15% of women completed the nutrition self-assessment.	Not reported	Not reported
Hillesund et al. (2016) [44]	4245 women attended the clinics during the inclusion period, of 1610 were eligible and 606 were recruited. Attrition was equally distributed among groups.	Not reported	Women in the intervention group reported reading food labels more often, and buying smaller packages of unhealthy foods.	The intervention group had higher overall diet score and favorable dietary behavior in 7 of 10 domains.
Jackson et al. (2011) [45]	Not reported	98% liked the program overall, 98% found it (very) easy to use, and 94% thought it was adequately private, yet 27% thought the program was too long. More participants liked the Educational Worksheet (97%), than the Video Doctor portion (82%).	Nutrition knowledge improved more in the intervention group, and participants more often discussed nutrition with providers.	There were statistically significant increases in intake of fruits and vegetables, whole grains, fish, avocado and nuts, and significant decreases in intake of sugary foods, refined grains, high-fat meats, fried foods, solid fats, and fast food.
Kafatos et al. (1989) [46]	Not reported	Not reported	Not reported	Energy and protein intakes were significantly closer to recommendations in the intervention group. There were improvements in concentrations of β-carotene and serum vitamin C, but not in hemoglobin, serum iron, and serum vitamin A.
Mauriello et al. (2016) [47]	Good engagement and retention. Nearly 100% of invited women participated, 70–77% of participants were retained at each time point.	Not reported	Significantly more intervention group participants progressed to the action or maintenance Stages of Changes.	There were no significant differences in intakes of fruits and vegetables during pregnancy.
Mauriello et al. (2011) [48]	Recruitment goals were met and exceeded within 3 weeks. All recruited women agreed to participate, 86% completed the session.	90–95% was very satisfied with the program. Participants liked learning new information (n = 35), tailored and personalized feedback (n = 9), and found the program easy to use (n = 6). Some thought there was too much repetition of questions (n = 9) or that the program took too long to complete (n = 6).	Improved assessment of advantages of changing behavior and intentions to change behavior.	Participants reported an average of 1.7 more servings of fruits and vegetables, each day post-intervention.
Moniz et al. (2015) [49]	Not reported	Not reported	Beliefs about nutritious foods and taking daily vitamins improved in 84% and 83% of participants, respectively.	41% of participant reported a higher frequency of nutritious food intake and 32% took vitamins supplements more often.
Piirainen et al. (2016) [50]	Not reported	215 women attended all study visits. The proportion of women who consumed the provided food products for each 12-week period between study visits ranged from 68% to 100%, depending on the product.	Not reported	Significantly higher intakes of vegetables, fruits, soft margarines, and vegetable oils and lower intake of butter. Higher intakes of PUFA, and lower intakes of SFA, as well as higher intakes of vitamin E, folate, and vitamin C.
Rissel et al. (2019) [51]	Severe issues with reach and uptake: 3736 women were screened, 923 found eligible, 322 enrolled, and only 89 completed the final call.	64% of women in the health-coaching arm received all 8 calls, 17% received 5–7 calls and 19% received ≤4 calls.	Not reported	No significant differences in serves of fruit and vegetables, cups of soft drinks, or frequency of take-away meals.
Warren et al., (2017) [52]	Not reported	Participants frequently referred back to their goal card. Authors report acceptability was very high. Women felt it helped to re-assess their eating behavior and think differently about their diet, and it gave them a sense of reassurance.	Not reported	All participants but one reported that the program improved the quality of their diet.
Wilkinson & McIntyre (2012) [53]	Approximately half (48.3%) of the intervention women attended the workshop and overall response rate at time 2 was 67.2%.	Not reported	Not reported	Significantly better adherence to fruit guidelines at time 2. Women who attended the workshop increased their consumption of serves of fruit, vegetables, met fruit guidelines, and had a higher diet quality score.
Wilkinson et al. (2010) [54]	Retention rates were lower in the intervention group (85.9%, 57.7%, and 49.1% at baseline, 12-weeks and 24-weeks post-service entry, respectively) compared to the control group (92.2%, 85.8%, and 75.2%)	Not reported	Not reported	No significant effect on fruit and vegetable intakes.

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
