# Peer review of "Dietary Interventions for Healthy Pregnant Women: A Systematic Review of Tools to Promote a Healthy Antenatal Dietary Intake"

_nutrients, 2020, doi:10.3390/nu12071981_

Round 1
Reviewer 1 Report
The manuscript by Beulen et al. entitled “ Dietary interventions for healthy pregnant women: A systematic review of tools to promote a healthy antenatal dietary intake” describes a systematic review in which available tools to promote a healthy dietary intake among healthy pregnant women in western countries are mapped. The authors have conducted a well prepared systematic review, in which they have gathered feasibility and effectiveness data of these tools. Attention to the comments below could further strengthen the manuscript.
Major comments:
1. The aim of this manuscript is to inform the development of a toolbox to promote a healthy dietary intake. However, in this manuscript the authors do not present the eventual toolbox, but only the previous steps: overview of tools on feasibility and effectiveness. It would make more sense to have that as aim for this review, with a practical application that the overview of these tools could help select health care providers or intervention developers in choosing the appropriate tool. Furthermore, the mention of a “toolbox” is misleading it might be better to relate to ‘an overview of tools’.
2. The authors use a wide definition of ‘tools’ based on the definition of McCalman (2014) and include tools “as any structured guideline, instrument, package or resource that supports antenatal care providers to integrate nutritional communication into their current practice.” Furthermore they include tools that are directed at pregnant women directly, such as resources and education materials.
2a. The current manuscript could be improved as the distinction between tools targeted to the healthcare professionals and tools targeted directly to pregnant women would be more clearly indicated and described and compared separately. In the current overview in the tables only the tools for the pregnant women are described and not for example training materials/manuals for the health care professional, which makes it sometimes unclear why a certain study was included.
2b. Secondly, I am not sure if vouchers for juice exchange or provision of foods can be considered as a tool based on the above definition (e.g. Article Burr et al. 2007 &Article Piirainen et al. 2016). This seems rather to be a financial incentive. A rational for why these are included should be provided.
3. The authors indicate they only include tools specific for a healthy pregnant population. Exclusion criteria comprise among others non-pregnant population or a diseased pregnant population. The diseased population is specified as e.g. women with Gesational diabetes mellitus or hyperemesis gravidarum. It would be helpful to specify here all excluded diseased populations with (i.e. ….). This is especially important as in the introduction section line 67-68 the authors specify they address two important gaps: 1. an overview of tools to promote a healthy dietary intake in pregnant women who are not visibly at risk. Pregnant women who are at risk in my opinion also comprise those overweight or obese, although it seems you include those tools among overweight / obese pregnant women. Regarding this point I am still confused. Perhaps the term healthy in ‘healthy pregnant women” should be chosen differently, such as “non-diseased pregnant women”. Especially since tools for those overweight and obese are especially valuable and most needed.
4. In light of the previous comments it would be valuable to provide beside SES information also other descriptive information such as weight or BMI.
5. Results: 3.3. Effectiveness paragraph. The authors only mention the positive effects of their included studies. However, only half of the included studies showed positive effect. It would be better to restructure this paragraph and give equal prominence to both the positive and negative findings perhaps clustered on specific outcome.
6. Results: 3.3.4. Line 344. The authors mention that the most intensive interventions were not necessarily the most effective ones. However, different outcome measures and different populations make this comparison impossible to make on the available data.
7. Discussion: Line 384-394: the authors refer specifically to the SES population, which was mainly included in 5 studies. It would help the reader if in the result section already some mention would be made regarding the results for this sub analysis on the SES population.
Minor comments:
- Introduction line 50: abbreviation HCP, please write in full the first time used.
- Methods 2.4 Line 139-140: the authors specify they considered for feasibility any outcome related to time, resources, facilitators and barriers for implementation. However in Table 1 the authors specify for feasibility outcomes: recruitment & retention (reach, response rate and retention rate and acceptability (satisfaction, dose received). Please change accordingly in the text, since for example no outcomes related to facilitators and barriers were presented.
- Methods 2.5 Line 147-148 & Line 202 (title) & Line 320 (title) & Line 356 & line 379: Perhaps use instead of contents “content”. All others were formulate as single as well.
- Methods 2.5 line 150: Perhaps use instead of full intervention “complete intervention”.
- Results 3.2.3 line288: Use instead of influence OF disrupt “Influence OR disrupt”
- Results 3.2.4 Line 291-292: Sentence seems incorrect and is difficult to understand. Should it be “multiple contacts help to spread time investment for providers across pregnancy”
- Results 3.3 Line 296-301: It is advised to combine the first sentence with the remainder sentences, such as: “dietary intake (i.e. food frequency questionnaires (n=?) [REFERENCES], (weighed) food records (n=?) [REFERENCES], 24h dietary recalls (n=?) [references] etcetera) and other health behaviors (i.e. ….) and/or determinants of these behaviors (i.e. ….). Although it remains a question why the authors mention these other behaviors here, because they were only interested in the effectiveness of dietary intake.
- Results 3.3 Line 303: the authors mention “dietary behaviour”, while previously referring to “dietary intake” (line 296). It is advised to use the same term throughout the manuscript.
- Results: the authors nowhere make a mention to table 4.
- Tables: in the tables in which you mention all articles it would be helpful to have the reference number, this makes it easier to look up studies, while reading the manuscript.
Reviewer 2 Report
General Feedback and Recommendations
- Overall, the work described in this review manuscript is timely and important. There are very few review articles that synthesize the literature on evidence based health promotion tools, their feasibility and effectiveness for pregnant women.
- This manuscript will be relevant and interesting to Nutrients readership because it gives a summative view on an area of healthcare and assessment that is unique and lacking in the literature. This review could provide basis for future programs and study design in relation to health promotion in pregnancy.
- Overall, the manuscript is well organized and clearly articulated. There are only a few very minor grammatical issues. I have a few comments on points to address listed below, but overall this is an excellent review.
- I recommend acceptance after very minor revisions.
Introduction
Overall: The introduction is very well organized and lays out a very clear statement of the problem.
- Page 2, Line 61: please spell out NCDs with the acronym in parentheses, as this is the first time it appears in the document.
Materials and Methods
Overall: the authors have clearly organized this section and provide justification and citations for methodological definitions and decisions that are sound and relevant to the field.
- Page 2, Lines 76-79: Did the researchers register the systematic review protocol prior to data collection? It is indicated as N/A per the Prisma checklist. As such, authors need to provide justification for this decision and provide slightly more detail in the methods on how the protocol was free of bias and reviewed by the librarian.
- Page 2, Lines 81- 82: How/why did authors select PubMed and Web of Science as the search databases? Provide a bit more detail on that choice.
- Page 3, Eligibility criteria: Did the researchers use a publication date range as part of the criteria? Ie. Only studies published in last 10 or 20 years?
- Page 21 and throughout: There are a few instances where sentences are not grammatically clear. Please review and revise throughout. Example: Line 291-292.
Discussion and Conclusions:
- Overall, the discussion and conclusion are very well organized and clearly articulated from the findings.
- Pending space: it would be interesting to see a bit more discussion about policy and practical applications in the health care setting and how to train or set up health professionals to more adequately support diet and health behavior change for pregnant women.
Round 2
Reviewer 1 Report
No further comments. The authors have addressed all previous issues correctly.